# DPO: Dual-Perturbation Optimization for Test-time Adaptation in 3D Object Detection

Zhuoxiao Chen*
The University of Queensland
Brisbane, Australia
zhuoxiao.chen@uq.edu.au

Zixin Wang*
The University of Queensland
Brisbane, Australia
zixin.wang@uq.edu.au

Yadan Luo†
The University of Queensland
Brisbane, Australia
y.luo@uq.edu.au

Sen Wang
The University of Queensland
Brisbane, Australia
sen.wang@uq.edu.au

Zi Huang
The University of Queensland
Brisbane, Australia
helen.huang@uq.edu.au

## Abstract

LiDAR-based 3D object detection has seen impressive advances in recent times. However, deploying trained 3D detectors in the real world often yields unsatisfactory performance when the distribution of the test data significantly deviates from the training data due to different weather conditions, object sizes, *etc.* A key factor in this performance degradation is the diminished generalizability of pre-trained models, which creates a sharp loss landscape during training. Such sharpness, when encountered during testing, can precipitate significant performance declines, even with minor data variations. To address the aforementioned challenges, we propose **dual-perturbation optimization (DPO)** for **Test-time Adaptation** in **3D Object Detection** **(TTA-3OD)**. We minimize the sharpness to cultivate a flat loss landscape to ensure model resiliency to minor data variations, thereby enhancing the generalization of the adaptation process. To fully capture the inherent variability of the test point clouds, we further introduce adversarial perturbation to the input BEV features to better simulate the noisy test environment. As the dual perturbation strategy relies on trustworthy supervision signals, we utilize a reliable Hungarian matcher to filter out pseudo-labels sensitive to perturbations. Additionally, we introduce early Hungarian cutoff to avoid error accumulation from incorrect pseudo-labels by halting the adaptation process. Extensive experiments across three types of transfer tasks demonstrate that the proposed DPO significantly surpasses previous state-of-the-art approaches, specifically on Waymo → KITTI, outperforming the most competitive baseline by 57.72% in $AP_{3D}$ and reaching 91% of the fully supervised upper bound. Our code is available at https://github.com/Jo-wang/DPO.

*Both authors contributed equally to this research.
†Corresponding author.

## CCS Concepts

• **Computing methodologies** → **Scene understanding**; **Online learning settings**; **Object detection**; **Transfer learning**.

## Keywords

Test-time Adaptation, 3D Object Detection.

**ACM Reference Format:**
Zhuoxiao Chen, Zixin Wang, Yadan Luo, Sen Wang, and Zi Huang. 2024. DPO: Dual-Perturbation Optimization for Test-time Adaptation in 3D Object Detection. In *Proceedings of the 32nd ACM International Conference on Multimedia (MM '24), October 28-November 1, 2024, Melbourne, VIC, Australia.* ACM, New York, NY, USA, 10 pages. https://doi.org/10.1145/3664647.3681040

## 1 Introduction

LiDAR-based 3D object detection has gained significant attention with the rapid advancements in autonomous driving [10, 25, 31, 37, 38, 48] and robotics [42, 84], where mainstream 3D detectors are developed to interpret pure point clouds or fuse multimodal knowledge, commonly incorporating camera images [8, 36, 64]. However, deploying either point clouds-based or multimodal 3D detection models in real-world scenarios often leads to performance degradation due to distribution shifts between the training data and the encountered real-world data. For instance, a 3D detector trained on the nuScenes dataset [2] might suffer a performance drop when applied to the KITTI dataset [14] due to variations in object sizes and the number of beams. This is known as **cross-dataset shift**. Additionally, the shift can arise from real-world disturbances, termed as **corruption-based shift** [11, 20, 28], which includes challenges like diverse weather conditions and sensor malfunctions. Moreover, multiple factors are likely to be concurrent, for instance, deploying a 3D detector in a different city while suffering severe snow. This scenario is termed as **composite domain shift**.

Domain adaption has been discovered [5, 39, 68, 69] to mitigate the performance gap brought by various domain shifts. In 3D object detection, this involves aligning features between the labeled training data and the shifted test data to learn a domain-invariant representation [40, 79, 81] or conducting self-training with the aid of selected pseudo-labels [6, 32, 47, 73, 74]. However, these approaches necessitate extensive training over multiple epochs on both training and test sets, rendering them impractical for adaptation to the streaming data. Moreover, the exposure of the training

data can significantly compromise its privacy, especially when it contains sensitive user information (*e.g.*, user vehicle trajectories and individuals).

To bridge the performance gap induced by domain shifts, while safeguarding the training data privacy and enabling swift adaptation, test-time adaptation (TTA) emerges as an ideal solution. Prior research on TTA typically adapts a source pre-trained model to the unlabeled test data, either through updating a selected subset of parameters (*e.g.*, BatchNorm layers) [46, 55, 63], or employing the mean-teacher model [58, 65, 66, 77] within a single epoch. However, these TTA works currently applied in image classification are inadequate for addressing the dual demands (*i.e.*, object localization and classification) for supervision signals inherent in detection tasks. Within this context, MemCLR [62] stands out by refining the Region of Interest (RoI) features of detected objects through a transformer-based memory module for 2D object detection. Nevertheless, the stored target representations derived from the source pre-trained model cause performance degradation due to distribution shifts. These limitations pose significant challenges in utilizing previous TTA techniques for 3D object detection.

To tackle these challenges, our goal is to devise an effective strategy for adapting the 3D detection model to various data shifts. We observe a common performance decline when the model encounters unfamiliar scenes. This degradation primarily occurs as the model tends to converge to sharp minima in the loss landscape during training [13]. Such convergence makes the model vulnerable to slight deviations in the test data, leading to a performance drop. Furthermore, high variability and limited availability of the test data significantly increase the vulnerability of the pre-trained source model. In response, we propose **DPO** to secure adaptation generalizability and robustness through a worst-case **D**ual-**P**erturbation **O**ptimization in both model weight and input spaces. Specifically, at the model level, we apply a **perturbation in the weight space** [13] to the model's parameters to maximize loss within a predefined range, thereby optimizing the model toward noise-tolerant flat minima. However, due to the notable discrepancies between the training and testing scenes, merely weight perturbation is insufficient to fully address the extensive variability and complexity encountered in the 3D testing scenes. To overcome this, we augment our approach by incorporating an **adversarial perturbation on the BEV feature** of the test sample via element-wise addition. Once the model is adapted to maintain stability despite perturbed inputs, it becomes more resilient to noisy data, thereby enhancing its robustness. The generalization and robustness of the adaptation model heavily rely on accurate supervision—that is, adapting the detection model based on reliable pseudo-labeled 3D boxes. The supervision signals offered in previous works are either too weak for 3D detection tasks [46, 63] or excessively dependent on pre-trained source models [62], which might be compromised by domain shifts. To this end, we introduce a **reliable Hungarian matcher** to ensure trustworthy pseudo-labels by filtering out 3D boxes that exhibit high matching costs before and after perturbations. The underlying assumption is that, given arbitrary perturbations, the prediction is more trustworthy if the model can still produce consistent box predictions. A consistently low Hungarian cost for pseudo-labels across recent test batches indicates the model has been sufficiently

robust to shifts/noise in the test domain. Hence, to preserve generalization and minimize unnecessary computational expenses, we propose an **Early Hungarian Cutoff** strategy based on the Hungarian costs. We apply a moving average of the cost values from the current and all previous batches to determine when to cease the adaptation. Our approach exhibits state-of-the-art results surpassing previous TTA methods. We summarize our key contributions as follows:

- We introduce TTA in LiDAR-based 3D object detection (TTA-3OD). To the best of our knowledge, this is the first work to adapt the 3D object detector during test time. To tackle the challenge in TTA-3OD, we prioritize the importance of model generalizability and reliable supervision.
- We propose a dual-perturbation optimization (**DPO**) mechanism, which maximizes the model perturbation and introduces input perturbation. This strategy is key to maintaining the model's generalizability and robustness during updates.
- We leverage a Hungarian matching algorithm to facilitate the selection of noise-insensitive pseudo-labels, to bolster adaptation performance through self-training. This further serves as a criterion for appropriately timing the cessation of model updates.
- By conducting thorough evaluations of DPO across various scenarios, including cross-domain, corruption-based, and notably complex composite domain shifts, our approach showcases outstanding performance in LiDAR-based 3D object detection tasks, specifically on Waymo $\rightarrow$ KITTI, outperforming the most competitive baseline by 57.72% in $AP_{3D}$, and achieve 91% of the fully supervised upper bound.

## 2 Related Work

### 2.1 Domain Adaptive 3D Object Detection

Adaptation for 3D Object Detection focuses on transferring knowledge from 3D detectors trained on labeled source point clouds to unlabeled target domains, effectively reducing the domain discrepancies across diverse 3D environments such as variations in object statistics [61, 67], weather conditions [20, 72], sensor differences [19, 49, 71], sensor failures [28], and the synthetic-to-real gap [9, 30, 50]. Strategies to overcome these challenges include adversarial feature alignment [81], 3D pseudo-labels [6, 7, 23, 33, 47, 51, 60, 73–75], the mean-teacher model [21, 40] for prediction consistency, and contrastive learning [79]. Nonetheless, these cross-domain adaptation methods typically necessitate adaptation over multiple epochs, making them less suited for real-time test scenarios.

### 2.2 Test-time Adaptation in 2D Vision Tasks

Test-time adaptation (TTA) [34, 70] is designed to address domain shifts between the training and testing data [70] during inference time. As a representative, Tent [63] leverages entropy minimization for BatchNorm adaptation. Subsequent works [16, 22, 43, 44, 53] such as EATA [45], identifies reliable and nonredundant samples to optimize. DUA [41] introduces adaptive momentum in a new normalization layer whereas RoTTA [77] and DELTA [83] leverage global statistics for batch norm updates. Furthermore, SoTTA [17] and SAR [46] improve BatchNorm optimization by minimizing the loss sharpness. Alternatively, some approaches optimize the

entire network through the mean-teacher framework for stable supervision [59, 65], generate reliable pseudo-labels for self-training [18, 78], employ feature clustering [4, 26, 66], and utilizing augmentations to enhance model robustness [80]. However, these TTA methods are developed for general image classification. Additionally, MemCLR [62] applies TTA for image-based 2D object detection, using a mean-teacher approach to align instance-level features. Nevertheless, the applicability of these image-based TTA methods to object detection from 3D point clouds remains unexplored.

## 2.3 Generalization through Flat Minima

The concept of flat minima has been demonstrated to enhance model generalization. A prime example is SAM [13], which improves generalization by simultaneously optimizing the original objective (*e.g.*, cross-entropy loss) and the flatness of the loss surface. Besides, ASAM [29] aligns the sharpness with the generalization gap by re-weighting the perturbation according to the normalization operator. To deal with the presence of multiple minima within the perturbation's reach, GSAM [85] minimizes the surrogate gap between the perturbed and the original loss to avoid sharp minima with low perturbed loss. Moreover, GAM [82] introduces first-order flatness, which controls the maximum gradient norm in the neighborhood of minima. Current research on flat minima focuses mainly on supervised learning. While in TTA, the effectiveness of these strategies significantly relies on supervision signals and the shift severity of the test data, which suggests that the anticipated advantages of flat minima might not consistently materialize as expected.

## 3 Method

### 3.1 Notations and Task Definition

Considering a neural network-based 3D object detector $f_S(\cdot; \Theta_S)$ parameterized by $\Theta_S$, which is pre-trained on a labeled training point clouds drawn from the source distribution $\mathcal{D}_S$, **Test-Time Adaptation for 3D Object Detection (TTA-3OD)** aims to adapt $f_S(\cdot; \Theta_S)$ to the unlabeled test point clouds $\{X_t\}_{t=1}^T \sim \mathcal{D}_{\mathcal{T}}$ during test time in a single pass. $\mathcal{D}_S \neq \mathcal{D}_{\mathcal{T}}$ as the test point clouds are shifted due to varied real-world conditions. Here, $X_t$ represents the $t$-th batch of test point clouds, with $f_t(\cdot; \Theta_t)$ indicating the 3D detection model adapted for the $t$-th batch.

**Challenges in TTA-3OD.** The primary challenges of TTA-3OD lie in two aspects: (1) adapting the 3D detection model to unfamiliar test scenes often generates large and noisy gradients, leading to an **unstable** adaptation process. This instability hampers the model's ability to generalize effectively to the target domain; (2) uncontrollable **variations** in the testing scenes, such as environmental changes or sensor inaccuracies, can significantly compromise the quality and integrity of 3D scenes. Consequently, models trained on clean datasets struggle to maintain effectiveness and robustness when facing such distorted data, drastically diminishing their adaptation performance.

To address the above two challenges, our method fundamentally enhances 1) the model's generalization and stability when adapting to new domains and 2) its robustness against noisy/corrupted data, by optimizing the ***sharpness*** of the loss landscape during model adaptation with the proposed dual-perturbation applied to both the model's weights and input data.

## 3.2 Minimizing Sharpness in the Weight Space

The *sharpness* of the training loss, is the rate of change in the surrounding region of the loss landscape. It has been identified to be empirically correlated with the generalization error [15, 24, 27]. Motivated by this, recent works propose to reduce the loss sharpness during the training phase, aiming to improve the generalization capabilities of the model. One notable example is Sharpness-Aware Minimization (SAM), which enhances model training by integrating and optimizing the worst-case perturbations in model weights. The fundamental principle of SAM is that by minimizing the loss with respect to maximally perturbed weights within a vicinity, the entire vicinity (*i.e.*, all losses within it) is minimized. This directs the optimization trajectory toward a *flat minima* in the loss landscape. A *flat minima* is indicative of superior generalization capabilities, as the loss over it is less sensitive to large perturbations and/or noise in the model weights, unlike *sharp minima*.

In the context of TTA-3OD, the loss *sharpness* [1] during the adaptation can be formally defined as follows:

**Definition 3.1** (Loss Sharpness). The *sharpness* of the loss $\mathcal{L}_{\det}(X_t; \Theta_t)$ is of 3D detection model $f_t(\cdot; \Theta_t)$ to test the $t$-th batch of target point cloud $X_t$, denoted as $s(\Theta_t, X_t)$, is given by

$$s(\Theta_t, X_t) \triangleq \max_{\|\epsilon\|_2 \leq \rho} \mathcal{L}_{\det}(X_t; \Theta_t + \epsilon) - \mathcal{L}_{\det}(X_t; \Theta_t). \quad (1)$$

Here, $\epsilon$ is a perturbation vector in the weight space such that its Euclidean norm is bounded by $\rho$.

Previous literature [12, 13, 29, 35, 82, 85] calculates the sharpness by the loss between model predictions $f_t(X_t)$ and its ground truth labels $Y_t$. While no supervision is available during test time, a soft loss [17, 46, 63] is commonly employed with selective supervision. Next, the optimization of the detection loss and its sharpness is defined as:

$$\min_{\Theta_t} \max_{\|\epsilon_w\|_2 \leq \rho} \mathcal{L}_{\det}(X_t; \hat{Y}_t; \Theta_t + \epsilon_w). \quad (2)$$

The inner optimization aims to find a perturbation $\epsilon_w$ on model weights $\Theta_t$ within a Euclidean ball of radius $\rho$ to maximize the detection loss $\mathcal{L}_{\det}$, which is calculated based on the generated pseudo-labels $\hat{Y}_t$. To obtain the worst-case $\epsilon_w$, we draw inspiration from [13] to approximate the inner optimization by the first-order Taylor expansion:

$$\begin{aligned} \epsilon_w^*(\Theta_t) &\triangleq \underset{\|\epsilon_w\|_2 \leq \rho}{\arg\max} \mathcal{L}_{\det}(X_t; \hat{Y}_t; \Theta_t + \epsilon_w) \\ &\approx \underset{\|\epsilon_w\|_2 \leq \rho}{\arg\max} \; \mathcal{L}_{\det}(X_t; \hat{Y}_t; \Theta_t) + \epsilon_w^\top \nabla_{\Theta_t} \mathcal{L}_{\det}(X_t; \hat{Y}_t; \Theta_t) \\ &= \underset{\|\epsilon_w\|_2 \leq \rho}{\arg\max} \; \epsilon_w^\top \nabla_{\Theta_t} \mathcal{L}_{\det}(X_t; \hat{Y}_t; \Theta_t). \end{aligned} \quad (3)$$

Then $\hat{\epsilon}_w(\Theta_t)$, which satisfies this approximation, is derived by resolving a dual norm problem:

$$\begin{aligned} \hat{\epsilon}_w(\Theta_t) = \rho &\times \text{sign}(\nabla_{\Theta_t} \mathcal{L}_{\det}(X_t; \hat{Y}_t; \Theta_t)) \\ &\times \frac{|\nabla_{\Theta_t} \mathcal{L}_{\det}(X_t; \hat{Y}_t; \Theta_t)|}{\|\nabla_{\Theta_t} \mathcal{L}_{\det}(X_t; \hat{Y}_t; \Theta_t)\|_2}. \end{aligned} \quad (4)$$

To expedite computation, we omit the second-order term.

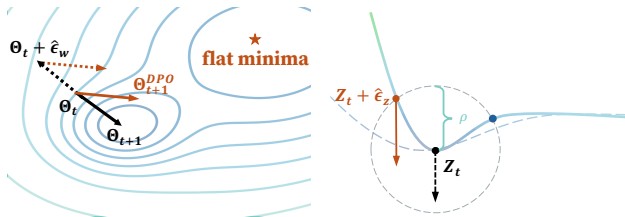

**Figure 1: (1) Loss contour for weight perturbation $\hat{\epsilon}_w$ (left); (2) The loss profile view for input perturbation $\hat{\epsilon}_z$ (right). Our goal is to optimize the loss towards flat minima while ensuring the model's resilience to data perturbations. Darker colors indicate lower loss values.**

While this improves model generalization by targeting non-sharp minima within the loss landscape, the optimized perturbations to weights do not directly deal with variations and/or noise of the input test scenes. When facing the test data, the detection performance is substantially degraded due to the severe data-level corruptions in the test point cloud. For example, when suffering heavy snow and shifted object scales, simultaneously, the performance drops from 73.45% to 3.84% in $AP_{3D}$. The empirical evidence suggests that augmenting the baseline model with SAM [46] results in a marginal improvement of only 0.9% in $AP_{3D}$, indicating its ineffectiveness in bridging the domain gap in the test 3D scenes.

### 3.3 Minimizing Sharpness in the Input Space

To surmount the above challenge, we strengthen the model's resilience against shifted input point clouds by optimizing the model with perturbed input. Rather than randomly mimicking test perturbations, our approach focuses on learning an adversarial perturbation that represents the worst-case corruption to the input data. The underlying rationale is that optimizing the detection model with maximal perturbed data within a given vicinity induces robustness to any perturbations encountered within that vicinity. As shown in **Figure 1**, we simultaneously guide the detection model toward the flat minima in both weight and input space, such that the model can stably generalize to the test data with any potential noises.

To introduce perturbations into the input batch, we incorporate an adversarial perturbation mask $\epsilon_z$ into the bird's eye view (BEV) feature map $Z_t$ through element-wise addition to each grid of the BEV map. This is because the 3D detector primarily localizes object proposals from the BEV map, which contains rich spatial information about 3D instances. Thus, even minimal perturbations to the feature map can cause significant spatial shifts in the instances, leading to misalignment in the final predicted 3D bounding boxes. To seek the worst-case perturbation $\epsilon_z$ within the input space that maximizes detection loss, we formulate the optimization problem as follows:

$$\epsilon_z^*(Z_t) \triangleq \underset{\|\epsilon_z\|_2 \leq \rho}{\arg\max} \ \mathcal{L}_{\det}(Z_t + \epsilon_z; \hat{Y}_t; \Theta_t). \tag{5}$$

Similar to approximating $\hat{\epsilon}_w(\Theta_t)$, we derive the approximated $\hat{\epsilon}_z(Z_t)$ within the input space. This resulting perturbation mask $\hat{\epsilon}_z$ shares the same dimension as the latent feature map $Z_t$ and is applied to $Z_t$ via element-wise addition, yielding the perturbed feature map $Z_t + \hat{\epsilon}_z$.

The final objective is to train the detection model with the optimal dual-perturbation in both model ($\hat{\epsilon}_w$) and input space ($\hat{\epsilon}_z$). To this end, we approximate the gradient by substituting $\hat{\epsilon}_w$ and $Z_t + \hat{\epsilon}_z$ into Eqn. (2), then performing differentiation to calculate the gradient $g$:

$$g = \nabla_{\Theta_t} \mathcal{L}_{\det}(Z_t + \hat{\epsilon}_z; \hat{Y}_t; \Theta_t)|_{\Theta_t + \hat{\epsilon}_w}. \tag{6}$$

Finally, the detection loss and its sharpness, calculated with the perturbed test batch, can be jointly minimized by:

$$\min_{\Theta_t} \max_{\substack{\|\epsilon_w\|_2 \leq \rho \\ \|\epsilon_z\|_2 \leq \rho}} \mathcal{L}_{\det}(Z_t + \epsilon_z; \hat{Y}_t; \Theta_t + \epsilon_w), \tag{7}$$

where the inner optimization is solved through approximation (*i.e.*, Eqn. (3)–(5)) and the outer optimization goal is achieved by stochastic gradient descent (SGD) with the gradient $g$ calculated in Eqn. (6). The step-by-step workflow of the proposed DPO is introduced in **Algorithm 1**.

### 3.4 Reliable Hungarian Matcher

However, the pursuit of flat minima in both the input and weight spaces depends on the gradients guided by high-quality supervision. Previous SAM-based TTA methods selectively adapt high-confidence samples [17, 46, 76], as they assume that confidence reflects prediction reliability. Nevertheless, acquiring effective supervision in the TTA-3OD task is challenging due to the low-quality pseudo-labeled boxes, $\hat{Y}_t = \{\hat{b}_1, \cdots, \hat{b}_{N_t}\}$, used for calculating the detection loss $\mathcal{L}_{\det}$, where $N_t$ represents the number of predicted boxes in the current batch $t$. This issue arises from the difficulties the source-trained model $f_S(\cdot; \Theta_S)$ faces in accurately predicting 3D boxes around objects in the test point clouds, which subjects to significant shifts or corruptions.

To obtain reliable pseudo-labeled boxes that are robust to the test data noise, we aim to select those 3D boxes **unaffected by optimized perturbations** (Sect. 3.3). The rationale is that consistency in box predictions between clean inputs and perturbed input features $Z_t + \hat{\epsilon}_z$ from the model before and after perturbation demonstrates resilience to noises $\hat{\epsilon}_z$. The box prediction from the $t$-th perturbed input batch is defined as:

$$\tilde{Y}_t = \{\tilde{b}_1, \cdots, \tilde{b}_{M_t}\} = f_t(Z_t + \hat{\epsilon}_z; \Theta_t + \hat{\epsilon}_w), \tag{8}$$

where $M_t$ is the number of predicted boxes at batch $t$ after perturbation. To measure the consistency between $\hat{Y}_t$ and $\tilde{Y}_t$, we adopt Hungarian matching [3, 54], an effective bipartite matching technique that guarantees optimal one-to-one alignment between two sets of box predictions. Specifically, We ensure both sets are of equal size by augmenting the smaller set (assuming $M_t < N_t$) with $\emptyset$ until it matches $N_t$ in size. To achieve optimal bipartite matching between the equal-sized sets, the Hungarian algorithm is applied to find a permutation of $N_t$ elements $p \in \mathbf{P}_{N_t}$ that minimizes the matching cost:

$$\tilde{p} = \underset{p \in \mathbf{P}_{N_t}}{\arg\min} \sum_n^{N_t} C_{\text{box}}(\hat{b}_n; \tilde{b}_{p(n)}). \tag{9}$$

The cost $C_{\text{box}}(\cdot; \cdot)$ integrates intersection-over-union (IoU) and L1 distance to account for the central coordinates, dimensions, and

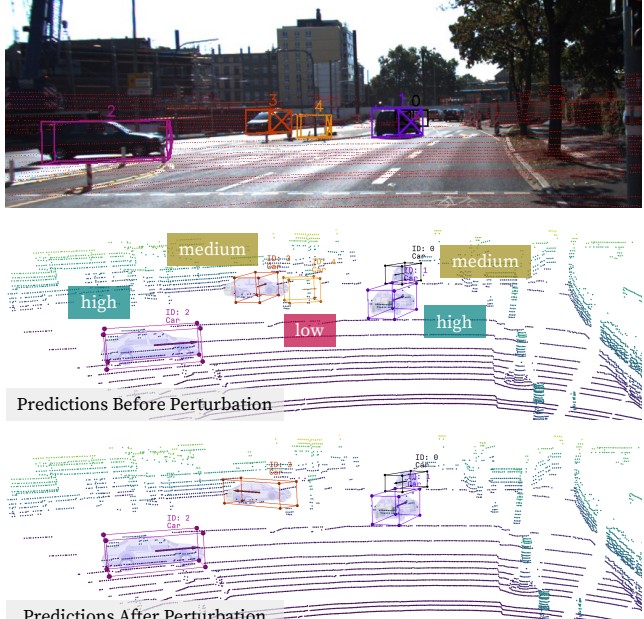

**Figure 2: Illustration of the proposed Hungarian matcher for obtaining reliable supervision. We employ the Hungarian algorithm to compute the cost for each pseudo-labeled 3D box (*i.e.*, predictions *before perturbation*) when paired with its optimally matched counterpart in predictions *after perturbation*. The reliability of the 3D boxes is categorized into three tiers—high, medium, and low—based on the computed matching cost. During TTA, only 3D boxes of high reliability (*e.g.*, ID 1, 2) are used for updating model weights, and those of low reliability (*e.g.*, ID 4) are treated as background.**

orientations of a pair of boxes: $\hat{b}_n$ and its corresponding matched box $\tilde{b}_{\tilde{p}(n)}$ indexed by $p(n)$.

Utilizing the derived optimal assignment $\tilde{p}$, each pseudo-labeled box $\hat{b}_n$ is associated with its corresponding minimal-cost match, denoted as $C_{\text{box}}(\hat{b}_n; \tilde{b}_{\tilde{p}(n)})$. Note that when $\hat{b}_n$ pairs with the empty set $\emptyset$, we assign the cost to be infinite, indicating that the corresponding box is too noise-sensitive to be accurately localized within the perturbed point clouds.

We categorize the reliability of each pseudo-label $\hat{b}_n$ into three distinct levels by the thresholds $C_1$ and $C_2$:

$$\begin{cases} \text{high} & \text{if } C_{\text{box}}(\hat{b}_n; \tilde{b}_{\tilde{p}(n)}) < C_1, \\ \text{low} & \text{if } C_{\text{box}}(\hat{b}_n; \tilde{b}_{\tilde{p}(n)}) > C_2, \\ \text{medium} & \text{otherwise.} \end{cases} \quad (10)$$

To guide the model towards flat minima with trustworthy supervision, we selectively adapt the model with high-quality bounding boxes and treat those of low quality as background, as shown in **Figure 2**. To dynamically set the thresholds $C_1$ and $C_2$, we record the minimum costs of pseudo-labeled boxes from previous batches in a sorted array $A_{\text{costs}}$, then determine $C_1$ and $C_2$ as the upper and lower $\alpha$ quantiles:

$$C_1 = A_{\text{costs}}[\lceil \alpha \times n \rceil], C_2 = A_{\text{costs}}[\lceil (1 - \alpha) \times n \rceil],$$
$$\text{where } A_{\text{costs}} = \text{sort}(\{C_{\text{box}}(\hat{b}_n; \tilde{b}_{\tilde{p}(n)})\}), \quad (11)$$
$$\hat{b}_n \in \{\hat{Y}_1\} \cup \cdots \{\hat{Y}_t\}, \tilde{b}_n \in \{\tilde{Y}_1\} \cup \cdots \{\tilde{Y}_t\}.$$

The ceiling function $\lceil \cdot \rceil$ ensures that the index for $A_{\text{costs}}$ is always an integer. Adopting global thresholds $C_1$ and $C_2$ derived from all historical costs facilitates more precise categorization of pseudo-labeled boxes into high and low-quality categories.

## 3.5 Early Hungarian Cutoff

While the Hungarian matcher significantly enhances the quality of pseudo-labels, the correctness of the selected pseudo-labels cannot be entirely guaranteed. Even a small number of incorrect pseudo-labels once learned and accumulated, can lead to substantial performance degradation. Furthermore, updating the 3D detector demands significant computational resources and time. Identifying an optimal stopping point for the adaptation process is thus crucial.

In this regard, we suggest using the Hungarian cost as a criterion to halt the adaptation process. The rationale is that a lower Hungarian cost for a given batch indicates the pseudo-labels are more accurate, thereby making the update process more reliable. Additionally, a consistently low Hungarian cost of pseudo-labels is crucial. Therefore, we introduce the use of a moving average to balance the current and all previous costs:

$$C_{\text{ema}}^t = \gamma C_{\text{box}}^t + (1 - \gamma) \sum C_{\text{box}}^{t-1},$$
$$\text{where } C_{\text{box}}^t = \frac{1}{N_t} \sum_n^{N_t} C_{\text{box}}(\hat{b}_n; \tilde{b}_{\tilde{p}(n)}), \hat{b}_n \in \hat{Y}_t, \tilde{b}_n \in \tilde{Y}_t, \quad (12)$$

is the average Hungarian cost of all boxes in the current batch $t$. $C_{\text{ema}}^t$ represents the moving average of the Hungarian cost, and $\gamma$ denotes the decay rate. A threshold $C_{\text{stop}}$ is further set for the moving average cost. When it falls below the threshold, the adaptation process is halted, the model thus transitions to the inference mode for all subsequent batches.

---

**Algorithm 1** DPO for TTA-3OD

**Input:** $f_S(\cdot; \Theta_S)$: source pre-trained model, $\{X_t\}_{t=1}^T \sim \mathcal{D}_\mathcal{T}$: target point clouds to test, $\eta$: step size, $C_{\text{stop}}$: early-stop threshold
**Output:** $f_t(\cdot; \Theta_t)$: model adapted to the target point clouds.
   Initiate the weights $\Theta_1 = \Theta_S$
   **for** $t = 1, \cdots, T$ **do**
      Generate predictions $\hat{Y}_t \leftarrow f_t(X_t; \Theta_t)$ as pseudo-label
      Compute perturbations $\hat{\epsilon}_z, \hat{\epsilon}_w$ via Eqn. (3)–(5)
      Generate prediction $\tilde{Y}_t$ with perturbations via Eqn. (8)
      Refine $\hat{Y}_t$ by reliable Hungarian matcher with $\tilde{Y}_t$ via (9)–(11)
      Compute gradient approximation $g$ via Eqn. (6)
      Update weights: $\Theta_{t+1} = \Theta_t - \eta g$
      /** check early stopping **/
      Compute the Hungarian matching cost $C_{\text{ema}}^t$ via Eqn. (12)
      **if** $C_{\text{ema}}^t \leq C_{\text{stop}}$ **then break**
         Infer the remaining batches with $f_t(X_t; \Theta_t)$
      **end if**
   **end for**

**Table 1: Results of test-time adaption to 3D scenes under cross-dataset shift. We report $AP_{BEV}$ / $AP_{3D}$ at moderate difficulty. Oracle means fully supervised training on the target dataset. The best adaptation results are highlighted in bold.**

| Method | Venue | TTA | Waymo →KITTI | | nuScenes →KITTI | |
|--------|-------|-----|--------------|--|-----------------|--|
| | | | $AP_{BEV}$ / $AP_{3D}$ | Closed Gap | $AP_{BEV}$ / $AP_{3D}$ | Closed Gap |
| No Adapt. | - | - | 67.64 / 27.48 | - | 51.84 / 17.92 | - |
| SN | CVPR'20 | ✗ | 78.96 / 59.20 | +72.33% / +69.00% | 40.03 / 21.23 | +37.55% / +5.96% |
| ST3D | CVPR'21 | ✗ | 82.19 / 61.83 | +92.97% / +74.72% | 75.94 / 54.13 | +76.63% / +65.21% |
| Oracle | - | - | 83.29 / 73.45 | - | 83.29 / 73.45 | - |
| Tent | ICLR'21 | ✓ | 65.09 / 30.12 | −16.29% / +5.74% | 46.90 / 18.83 | −15.71% / +1.64% |
| CoTTA | CVPR'22 | ✓ | 67.46 / 35.34 | −1.15% / +17.10% | 68.81 / 47.61 | +53.96%/ +53.47% |
| SAR | ICLR'23 | ✓ | 65.81 / 30.39 | −11.69% / +6.33% | 61.34 / 35.74 | +30.21% / +32.09% |
| MemCLR | WACV'23 | ✓ | 65.61 / 29.83 | −12.97% / +5.11% | 61.47 / 35.76 | +30.62% / +32.13% |
| **DPO** | - | ✓ | **75.81 / 55.74** | **+52.20% / +61.47%** | **73.27 / 54.38** | **+68.13%/+65.66%** |

## 4 Experiments

### 4.1 Experimental Setup

*4.1.1 Datasets and TTA-3OD Tasks.* Our experiments are carried out on three widely used LiDAR-based 3D object detection datasets: KITTI [14], Waymo [56], and nuScenes [2]. Additionally, the recently released KITTI-C dataset [28], which simulates real-world corruptions, is incorporated to pose the TTA-3OD challenge. Following prior works [6, 73, 74], we address **cross-dataset** test-time adaptation tasks (*e.g.,* Waymo → KITTI and nuScenes → KITTI), involving adaptation across (i) object shifts (*e.g.,* scale and point density variations), and (ii) environmental shifts (*e.g.,* changes in deployment locations and LiDAR configurations). Furthermore, we evaluate adaptation performance against **real-world corruptions** (*e.g.,* KITTI → KITTI-C), including conditions such as fog, wet conditions (Wet.), snow, motion blur (Moti.), missing beams (Beam.), crosstalk (Cross.T), incomplete echoes (Inc.), and cross-sensor interference (Cross.S). Experiments also extend to the challenging scenarios of **composite domain shifts** (*e.g.,* Waymo → KITTI-C), where inconsistencies across datasets and corruptions coexist within the test 3D scenes.

*4.1.2 Implementation Details.* We leverage the OpenPCDet framework [57]. Experiments are conducted on a single NVIDIA RTX A6000 GPU with 48 GB of memory. We opt for a batch size of 8 and fix the hyperparameters $\rho = 1e-4, \alpha = 0.08, \gamma = 0.5, \eta = 10^{-3}$. For evaluation purposes, we adhere to the official metrics of the KITTI benchmark, reporting the average precision for the car class in both 3D (*i.e.,* $AP_{3D}$) and bird's eye view (*i.e.,* $AP_{BEV}$) perspectives, calculated over 40 recall positions and applying a 0.7 IoU threshold. The closed gap [73] is calculated as: $\frac{AP_{method} - AP_{No\ Adapt.}}{AP_{Oracle} - AP_{No\ Adapt.}} \times 100\%$.

*4.1.3 Baseline Methods.* We integrate a voxel-based backbone (*i.e.,* SECOND) into our proposed method and evaluate it against a comprehensive array of baseline approaches:
- **No Adapt.**: directly inferring the test data with a model pretrained on the source domain, without any adaptation.
- **SN** [67]: weakly supervised domain adaptive 3D detection that adjusts source object sizes using target domain statistics.
- **ST3D** [73]: an unsupervised domain adaptation method for 3D detection, utilizing multi-epoch pseudo-labeling for self-training.
- **Tent** [63]: a fully TTA method that optimizes BatchNorm layers by minimizing the entropy of predictions.

- **CoTTA** [65]: a TTA strategy that leverages mean-teacher framework to provide supervisory signals through augmentations and employs random neuron restoration to retain source knowledge.
- **SAR** [46]: an advancement beyond Tent, employing sharpness-aware minimization for selected supervision.
- **MemCLR** [62]: TTA for *image-based object detection* using mean-teacher to align the instance-level features by a memory module.
- **Oracle**: a *fully supervised* model trained on the test scenes.

### 4.2 Experimental Results

*4.2.1 Cross-dataset Shifts.* We conducted extensive experiments on two cross-dataset TTA-3OD tasks, evaluating $AP_{BEV}$, $AP_{3D}$, and closed gap, as presented in **Table 1**. Compared to direct inference (*i.e.* No Adapt.), our experiments revealed that existing TTA baselines might negatively impact adaptation in 3D object detection especially on $AP_{BEV}$ for the Waymo → KITTI task, indicating the importance of tailoring a TTA method specifically for 3D detection tasks. Additionally, compared to the most competitive baseline, CoTTA, DPO achieves significant improvements in $AP_{3D}$, with increases of 57.7 and 14.2% for the Waymo → KITTI and nuScenes → KITTI tasks, respectively. Similarly, DPO significantly outperforms CoTTA in $AP_{BEV}$, demonstrating a considerable margin. Besides, DPO effectively reduces the closed gap, demonstrating a closure of about 61.47 and 65.66% for the Waymo → KITTI and nuScenes → KITTI tasks, correspondingly, in $AP_{3D}$. Moreover, it achieves up to 91% and 87.5% of the fully supervised Oracle's performance in $AP_{BEV}$ for the respective tasks. Overall, our proposed DPO not only surpasses all TTA baselines but also delivers performances competitive with those of Unsupervised Domain Adaptation (UDA) and fully supervised learning, highlighting its effectiveness in bridging domain gaps in 3D object detection.

*4.2.2 Corruption Shifts.* We evaluated DPO's efficacy against corruption-induced shifts on KITTI → KITTI-C with *heavy* severity of eight real-world corruption by $AP_{3D}$ in *hard* difficulty scenarios. As indicated in **Table 2**, DPO outperforms all TTA baselines in terms of Mean $AP_{3D}$, exceeding the performance of the closest competitive baseline, Tent, by 1.2%. DPO consistently achieves top performance across most corruption types, demonstrating the enhanced robustness of DPO and its effectiveness in adapting 3D models to a wide array of corrupted environments.

**Table 2: Results of KITTI → KITTI-C on heavy corruptions.**

|        | No Adapt. | Tent  | CoTTA | SAR   | MemCLR | **DPO** |
|--------|-----------|-------|-------|-------|--------|---------|
| Fog    | 68.23     | 68.73 | 68.49 | 68.14 | 68.23  | **68.72** |
| Wet.   | 76.25     | 76.36 | 76.43 | 76.23 | 76.25  | **76.89** |
| Snow   | 59.07     | 59.50 | 59.45 | 58.78 | 58.74  | **60.80** |
| Moti.  | 38.21     | 38.15 | 38.62 | 38.12 | 37.57  | **38.71** |
| Beam.  | 53.93     | 53.85 | 53.98 | 53.75 | 53.49  | **54.06** |
| CrossT.| 75.49     | 74.67 | 72.22 | 74.51 | 74.25  | **75.52** |
| Inc.   | 25.68     | 26.44 | 27.35 | 26.42 | **27.47** | 27.16 |
| CrossS.| 41.08     | 41.17 | 40.80 | 40.63 | 40.90  | **42.09** |
| Mean   | 54.74     | 54.86 | 54.67 | 54.57 | 54.61  | **55.49** |

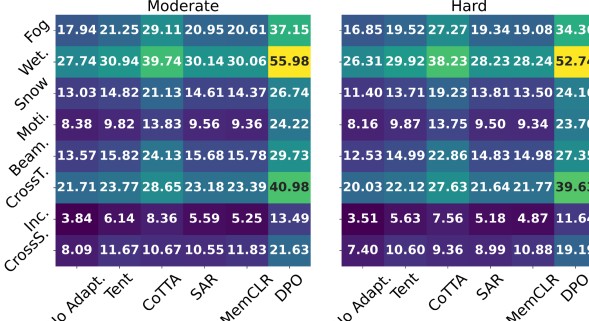

**Figure 3: Results (AP$_{3D}$) of adapting across composite shifts (Waymo → KITTI-C) at the heavy corruption level. Lighter shades indicate higher performance.**

*4.2.3 Composite Domain Shifts.* To address the most challenging shift in 3D scenes, which merges both cross-dataset discrepancies and corruptions, we conducted experiments to adapt 3D detectors from Waymo to KITTI-C (*heavy* corruption). The outcomes are represented in **Figure 3**. Notably, the shades in the last column (DPO) are significantly lighter than those in all other columns (TTA baselines) at various difficulty levels (*moderate* and *hard*), indicating DPO's superior performance. In particular, the performance without any adaptation (column 1) significantly declines, illustrating the compounded challenges of composite shifts. For example, only 8.38% AP$_{3D}$ is recorded for Motion Blur and 3.84% AP$_{3D}$ for Incomplete Echoes at the moderate level. Conversely, against the most competitive baseline (column 3), our approach notably enhances adaptation performance for these challenging corruptions by **75.13%** and **61.36%**, respectively. Direct inference for Incomplete Echoes at hard difficulty yields only a 3.51% in AP$_{3D}$, whereas our method markedly increases this by more than 231.62%, achieving a 53.97% improvement over the highest baseline. In summary, existing TTA methods fall short in navigating significant domain shifts (*i.e.*, composite domain shifts) in 3D scenes, while DPO could effectively tackle these challenges.

## 4.3 Parameter Sensitivity

*4.3.1 Sharpness Radius ρ.* To understand the impact of varying the sharpness radius $\rho$ on AP$_{3D}$ and AP$_{BEV}$, we conduct an analysis at the moderate difficulty level for the **nuScenes → KITTI** task, keeping all other hyperparameters fixed. We explored a range of $\rho$ values from $10^{-4}$ to $10^{-1}$. The left part of **Figure 4** illustrates

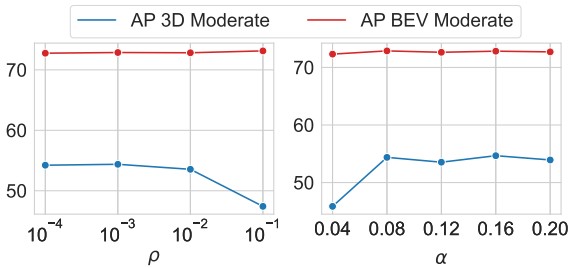

**Figure 4: Sensitivity to radius $\rho$ in SAM (left), and the pseudo-label threshold $\alpha$ (right) on nuScenes → KITTI.**

**Table 3: Ablation study on the nuScenes → KITTI task. AP$_{BEV}$ (left) and AP$_{3D}$ (right) (%) are reported for three levels of difficulty. The best results are highlighted in bold.**

| Pert. $\Theta_t$ | Pert. $Z_t$ | Matcher | Easy | Moderate | Hard |
|-----------|-----------|---------|------|----------|------|
| -         | -         | -       | 76.51/58.78 | 62.68/43.64 | 59.93/39.87 |
| √         | -         | -       | 82.14/56.36 | 70.86/47.18 | 68.91/44.62 |
| √         | √         | -       | 80.42/60.50 | 72.42/49.28 | 70.77/46.20 |
| √         | -         | √       | 81.08/62.88 | 73.09/49.60 | 71.86/47.19 |
| √         | √         | √       | **83.11/66.19** | **73.27/54.38** | **72.21/52.66** |

that variations in $\rho$ exhibit a minimal influence on AP$_{BEV}$, contrasting with AP$_{3D}$, which demonstrates significant performance variability when the perturbation radius is adjusted to 0.1. This discrepancy can be attributed to two primary factors. Firstly, an increase in perturbation radius adversely affects adaptation performance. Secondly, a larger perturbation radius results in the selection of a reduced number of pseudo-labeled 3D boxes for self-training due to the increased divergence in model predictions. However, when employing a perturbation radius within a smaller range (*e.g.*, $10^{-4}$-$10^{-2}$), the stability of AP$_{3D}$ is notably enhanced.

*4.3.2 Pseudo-label Threshold $\alpha$.* The pseudo-label threshold $\alpha$ shows a consistent pattern for AP$_{BEV}$, remaining stable across different values. However, a low threshold (*i.e.*, 0.04) causes a drop in AP$_{3D}$ as too few pseudo-labeled 3D boxes are selected to update model weights. This emphasizes the need for an appropriate proportion of pseudo-labels for adaptation. For $\alpha$ values between 0.08 and 0.20, AP$_{BEV}$ and AP$_{3D}$ remain stable, with maximum fluctuations of 0.83 and 0.07, respectively. This stability highlights the robustness of the selected threshold.

## 4.4 Ablation Study

*4.4.1 Impact of Components.* To understand how individual components of DPO affect overall performance, we conduct an ablation study by incrementally adding each component to adaptation and evaluating performance on the nuScenes → KITTI task. **Table 3** shows the impact of these components on the KITTI dataset at three difficulty levels, measured by AP score. Here, Pert. $\Theta_t$ represents weight space perturbation, Pert. $Z_t$ denotes input perturbation, and Matcher refers to the Hungarian Matching mechanism for pseudo-label selection. Compared to the self-training baseline (row 1), adding weight space perturbation (row 2) significantly improves AP$_{BEV}$ but reduces AP$_{3D}$ (58.78 → 56.36 at easy difficulty),

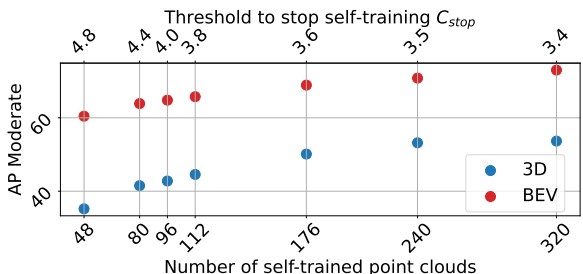

Figure 5: Performance trend and variation in the number of point clouds for weight updates across different early-stopping thresholds $C_{stop}$.

Table 4: DPO variations for model updating strategies.

|  | # of Params | Speed | $AP_{BEV}$ | $AP_{3D}$ |
|---|---|---|---|---|
| Full | 12,182,565 | 0.33s / frame | **75.81** | **55.74** |
| BatchNorm | **268,288** | **0.31s / frame** | 75.12 | 55.51 |

Table 5: DPO results of Waymo → KITTI using PV-RCNN.

| TTA Method | No Adapt. | Tent | CoTTA | SAR | Mem-CLR | Ours |
|---|---|---|---|---|---|---|
| $AP_{BEV}$ | 63.60 | 55.96 | 67.85 | 59.77 | 55.92 | **68.45** |
| $AP_{3D}$ | 22.01 | 27.49 | 38.52 | 21.33 | 15.77 | **51.55** |

indicating limitations of SAM for the TTA-3OD task. Incorporating input perturbation (row 3) and using dual perturbation improves performance, increasing $AP_{BEV}$ and $AP_{3D}$ across all difficulty levels. The Hungarian matcher enhances pseudo-label selection with weight perturbation alone, as shown by the performance gains over weight perturbation alone (rows 2, 4). Using all proposed DPO components yields the highest performance for both $AP_{BEV}$ and $AP_{3D}$ across all difficulty levels.

*4.4.2   **Impact of Early Hungarian Cutoff.***  We examine the effectiveness of early Hungarian cutoff on the nuScenes → Waymo task in **Figure 5**. When the Hungarian cost in Eqn. (12) falls below a specified threshold, *e.g.*, $C_{stop} = 4.8$, the model updates its weights using self-training on the first 48 test point clouds and then infers the remaining point clouds directly, skipping further self-training. The moving-average Hungarian cost converges rapidly during self-training. For instance, using the first 64 test samples reduces the cost from 4.8 to 3.8 and significantly improves performance (9.37 in $AP_{3D}$). In contrast, reducing the cost from 3.6 to 3.4 with 144 test samples only marginally improves performance (3.53 in $AP_{3D}$) due to error accumulation in pseudo-labels. These results underscore the value of the Hungarian cost-based early stopping mechanism, which leverages a small portion of test batches to enhance performance without excessive computational cost.

*4.4.3   **Impact of Updating Strategies.***  We explore updating only the BatchNorm (BN) vs. the full model for adaptation on Waymo → KITTI. As shown in **Table 4**, updating BN (only 2% of the total parameters) results in a slight decrease of 0.41% in $AP_{3D}$ and a slight increase in speed by 0.02s per frame. This demonstrates that our method remains effective even when only a small fraction of the parameters are updated.

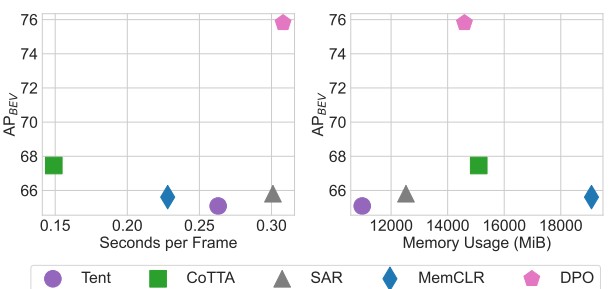

Figure 6: Efficiency analysis of Waymo → KITTI task.

*4.4.4   **Sensitivity to 3D Backbone Detector.***  To validate the effectiveness of DPO, we assess the performance sensitivity when coupled with a two-stage, point- and voxel-based backbone detector: PVRCNN [52]. The results of TTA baselines and our approach from Waymo to KITTI are summarized in **Table 5**. Our observations indicate that changing the backbone has a significant impact on the performance of baseline TTA methods. Conversely, our proposed method not only exhibits stability but also achieves a remarkable performance enhancement (33.83% in $AP_{3D}$) over the leading baseline. Besides, DPO also achieves state-of-the-art $AP_{BEV}$ performance compared to all baseline methods, emphasizing the consistent efficacy of our approach across different backbones.

## 4.5   Efficiency Analysis

To assess the efficiency of DPO, we conducted a comparative analysis for adaptation speed (*i.e.*, seconds per frame) and GPU memory usage, as illustrated in **Figure 6**. CoTTA is identified as the most efficient TTA baseline for 3D object detection, demonstrating rapid adaptation capabilities (under 0.15 seconds per frame). Conversely, other baselines, notably SAR, required significantly more adaptation time and yet underperformed, achieving $AP_{BEV}$ of less than 66%. Despite a slightly longer processing time, DPO markedly surpassed all TTA baselines, showcasing its superior performance. In terms of GPU memory consumption, CoTTA reported moderate usage, whereas MemCLR exceeded 18,000 MiB but fell short in performance. The proposed DPO, in contrast, not only required less GPU memory than both MemCLR and CoTTA but also achieved dominating adaptation performance, highlighting the efficiency and effectiveness of our method.

## 5   Conclusion

In this work, we present a novel framework for Test-Time Adaptation in 3D Object Detection (TTA-3OD) aimed at adapting detectors to new unlabeled scenes with a single pass. Our approach incorporates worst-case perturbations at both model and input levels to enhance robustness and generalization, thereby enabling 3D detectors to stably adapt to any test scenes with corruptions. We employ reliable Hungarian matching for trustworthy pseudo-label selection, with an early cutoff to avoid computation burden and error accumulation. Beyond point- and voxel-representation-based 3D detectors used in this paper, our future work will further validate multimodal detectors with different input modalities, such as BEVfusion [36] to verify shifts across modalities.

## Acknowledgments

This research is partially supported by the Australian Research Council (DE240100105, DP240101814, DP230101196, DP230101753).

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
