# OpenReview forum: "DPO: Dual-Perturbation Optimization for Test-time Adaptation in 3D Object Detection"
_acmmm.org/ACMMM/2024/Conference — MM2024 Poster_

### Official Review · Reviewer_KFiF · 2024-05-23

**Rating:** 4
**Confidence:** 2

**Summary:**

This paper proposes a dual-perturbation optimization method, called TTA-3OD, to improve 3D object detection models' test-time adaptation ability. On the one hand, TTA-3OD proposes to utilize weight and input perturbation to minimize the sharpness of data. This could improve the generalization and robustness of the model. Then, a Hungarian Matcher and early cutoff are adopted to predict precise pseudo-labels. The experiment result is promising.

**Strengths:**

1. The paper is well-written and easy to follow.
2. The problem is practical and the motivation is strong;
3. The experiment is well-designed and extensive.

**Limitations:**

1. In Table 3, there is a decrease in AP_{BEV} in terms of Easy difficulty from row 2 to row 3, could you please analyze the potential reason?
2. Despite the high test-time efficiency of TTA-3OD, could you please also analyze the training efficiency?

**Suitability:**

3

---

### Official Review · Reviewer_x1VS · 2024-05-24

**Rating:** 5
**Confidence:** 2

**Summary:**

This paper introduces the method to enhance the adaptability and robustness of LiDAR-based 3D object detection models in real-world scenarios. This method addresses performance degradation due to distribution shifts between training and test data by implementing a dual-perturbation strategy. The dual-perturbation optimization (DPO) method introduces perturbations in both the model weights and input data to achieve a flatter loss landscape, enhancing the model's resilience to data variations. Additionally, the method employs a reliable Hungarian matcher to filter out pseudo-labels sensitive to perturbations, ensuring the use of high-quality supervision signals during adaptation. Extensive experiments demonstrate that DPO significantly outperforms existing state-of-the-art methods in various transfer tasks.

**Strengths:**

Innovative Dual-Perturbation Strategy: The introduction of dual-perturbation optimization, which applies perturbations to both model weights and input data, is an innovative approach that significantly enhances the model's robustness and adaptability to data variations and noise.

High-Quality Supervision Mechanism: The use of a reliable Hungarian matcher to filter pseudo-labels ensures high-quality supervision during the adaptation process. This mechanism effectively mitigates the impact of noisy or incorrect labels, leading to more stable and accurate model updates.

Comprehensive Evaluation: The paper provides extensive experimental evaluations across various scenarios, including cross-dataset, corruption-based, and composite domain shifts. The results highlight the superior performance of DPO compared to existing methods, demonstrating its effectiveness in real-world applications.

**Limitations:**

Computational Overhead: The dual-perturbation strategy and Hungarian matching mechanism, while effective, may incur significant computational overhead. This could limit the practical applicability of the method in real-time or resource-limited settings.

Sensitivity to Hyperparameters: The method's performance is sensitive to hyperparameters, such as the perturbation radius and pseudo-label thresholds. Fine-tuning these parameters requires careful consideration and may not be straightforward, potentially hindering the method's ease of use and adaptability.

**Suitability:**

2

---

### Official Review · Reviewer_esSs · 2024-05-25

**Rating:** 3
**Confidence:** 4

**Summary:**

This paper proposes a approach to improve the performance of 3D object detectors during test-time adaptation (TTA). The method addresses the challenges posed by distribution shifts between training and test data. The proposed Dual-Perturbation Optimization (DPO) strategy aims to create a flat loss landscape to enhance the model's generalization and robustness to data variations. The experimental results demonstrate significant improvements over state-of-the-art approaches in various transfer tasks, including cross-dataset, corruption-based, and composite domain shifts.

**Strengths:**

1.The dual-perturbation approach enhances model robustness by addressing both weight and input feature spaces, ensuring improved generalization and stability.
2.The use of a Hungarian matcher to filter unreliable pseudo-labels ensures high-quality supervision. The early Hungarian cutoff strategy minimizes computational overhead while maintaining high adaptation performance.

**Limitations:**

1.This paper is well-structured, and the method is easy to follow, but some of the wording is obscure and needs further polish.
2.While this paper is the first to use test-time adaptation for 3D object detection, adding perturbation/data augmentation is already common in the UDA field. Therefore, the innovation is not very clear, and the contribution should be highlighted with more analysis and comparison.
3.Section 3.2 repeats too much of [11] and should highlight this paper's unique aspects. Also, the technical ideas in sections 3.2 and 3.3 are very similar to [11]. Is there any unique design specifically for the 3D object detection task?
4.This paper has too many hyperparameters and thresholds, like Sharpness Radius, Pseudo-label Threshold, C_1, C_2, and C_stop. This problematic for DA method, as it requires careful tuning for different tasks and datasets.

**Suitability:**

2

---

### Official Review · Reviewer_prRf · 2024-05-27

**Rating:** 3
**Confidence:** 4

**Summary:**

The paper introduces a new task, namely test-time Adaptation in 3D object detection and proposes dual-perturbation optimization (DPO). DPO add perturbation in weight and input space and ensure the model resiliency to minor data variations. Further, to obtain accurate pseudo labels, the authors propose a early Hungarian cutoff strategy. Experiments show that DPO can largely outperform previous 2D TTA works in 3D object detection task in different settings.

**Strengths:**

-	The method that add perturbation and ensure the stability of model is reasonable.

-	The paper is easy to follow.

**Limitations:**

-	The authors should report the number of updated parameters since the efficiency is important in TTA tasks. Besides, the authors add perturbation in weight and bev space, does it mean that a single frame will be calculated three times during adaptation? If so, I think the efficiency of the method need to be improved.

-	I wonder the result of combining DPO and more SOTA detectors such as DSVT. I notice that the results of using PV-RCNN are worse than use SECOND. However, PV-RCNN is a more powerful detector, I think the authors should give some explanation.

-	The authors should also conduct experiments under waymo-to-nuScenes settings.

-	More comparison should be included such as TTA in 3D [A][B].

[A] Mirza M J, Shin I, Lin W, et al. Mate: Masked autoencoders are online 3d test-time learners[C]//Proceedings of the IEEE/CVF International Conference on Computer Vision. 2023: 16709-16718.

[B] Wang Y, Cheraghian A, Hayder Z, et al. Backpropagation-free Network for 3D Test-time Adaptation[J]. arXiv preprint arXiv:2403.18442, 2024.

**Suitability:**

3

---

### Meta-Review · Area_Chair_8Z42 · 2024-07-01

**Recommendation:** Accept (Poster)
**Confidence:** 4

**Metareview:**

The initial ratings of this paper were mixed. However, after the rebuttal, the authors effectively addressed the concerns of most reviewers, resulting in all positive ratings. Therefore, this paper is recommend acceptance.